# High-Precision Coal Mine Microseismic P-Wave Arrival Picking via Physics-Constrained Deep Learning

**DOI:** 10.3390/s25237103

**Published:** 2025-11-21

**Authors:** Kai Qin, Zhigang Deng, Xiaohan Li, Zewei Lian, Jinjiao Ye

**Affiliations:** 1China Coal Research Institute, Beijing 100013, China; qinkai@ccri.com.cn (K.Q.); lixiaohan@ccri.com.cn (X.L.); lzw2000@alu.cau.edu.cn (Z.L.); yejinjiao@ccri.com.cn (J.Y.); 2State Key Laboratory of Coal Mine Disaster Prevention and Control, Beijing 100013, China

**Keywords:** microseismic monitoring, event detection, arrival time retrieval, deep learning

## Abstract

The automatic identification of P-wave arrival times in microseismic signals is crucial for the intelligent monitoring and early warning of dynamic hazards in coal mines. Traditional methods suffer from low accuracy and poor stability due to complex underground geological conditions and substantial noise interference. This paper proposes a microseismic P-wave arrival time automatic picking model that integrates physical constraints with a deep learning architecture. This study trained and optimized the model using a high-quality, manually labeled dataset. A systematic comparison with the AR picker algorithm and the short-term–long-term average ratio method revealed that this model achieved a precision of 96.60%, a recall of 90.59%, and an F1 score of 93.50% on the test set, with a P-wave arrival time-picking error of less than 20 ms. The average arrival time error was only 5.49 ms, significantly outperforming traditional methods. In cross-mining area generalization tests, the model performed excellently in two mining areas with consistent sampling frequencies (1000 Hz) and high signal-to-noise ratios, demonstrating good engineering transferability. However, its performance decreased in a mining area with a higher sampling rate and stronger noise, indicating its sensitivity to data acquisition parameters. This study developed a high-precision, robust, and potentially cross-domain adaptive model for automatically picking microseismic P-wave arrival times. This model provides support for the automation, precision, and intelligence of coal mine microseismic monitoring systems and has significant practical value in promoting real-time early warning and risk prevention for mine dynamic hazards.

## 1. Introduction

Microseismic monitoring, a key technique in modern geophysics, has been widely applied in deep mineral resource extraction, hydraulic fracturing surveillance in oil and gas fields, and early warning systems for significant geological hazards [1,2,3,4]. The method involves deploying high-sensitivity, broadband geophone arrays in underground rock masses to capture subtle seismic signals generated via rock fractures or slips. By inverting these signals, researchers can determine the source location, origin time, magnitude, and rupture mechanism, thereby dynamically characterizing the evolution of subsurface structures [5]. Accurately identifying the P-wave first arrival time is the most fundamental and critical step in this monitoring workflow. Its precision directly affects the reliability and resolution of subsequent source location inversions [6]. Particularly in complex engineering environments, such as deep coal mining and shale gas fracturing, microseismic events often have low magnitudes, short durations, and rapid energy decay. They are also prone to interference from strong background mechanical noise, environmental vibrations, and overlapping waveforms from multiple events. This interference causes P-wave onsets to be gradual, lacking a sharp amplitude jump, which poses a significant challenge for identifying the arrival time of microseismic events.

Researchers have proposed various automatic picking algorithms based on signal processing and statistical models. Early methods often focused on the differences between seismic signals and background noise regarding time-domain amplitude, frequency-domain characteristics, or statistical distribution, using a discriminant function to identify the first arrival time. Among these, the Short-Term-Long-Term Average Ratio (STA/LTA) method [7] and its derivatives have become some of the most widely used classic methods in the field due to their simple structure, computational efficiency, and ease of engineering deployment. Other researchers have combined these methods to improve accuracy. Ma et al. [8] and Jiang et al. [9] improved picking precision by integrating the AIC algorithm. Zhao et al. [10] proposed a method based on quality optimization and normalized STA/LTA, while Li et al. [11] introduced a feature function that reflects both signal energy and dispersion.

Additionally, methods like the Autoregressive Model combined with the Akaike Information Criterion (AR-AIC) [12,13], time-window energy feature methods [14], and higher-order statistical methods based on skewness and kurtosis [15,16] have also been extensively explored. While these algorithms perform well under ideal conditions, their effectiveness is severely limited in real-world, complex scenarios. Due to the non-stationary nature of microseismic waveforms and the effect of overlapping events, a single or limited-dimensional feature is often insufficient to capture the dynamic evolution of the signal. This leads to systemic flaws, particularly in low signal-to-noise ratio (SNR) conditions, including an increased rate of false positives, missed events, and excessive arrival-time deviations.

Groundbreaking advances in artificial intelligence, machine learning, and deep learning have gradually been integrated into the field of seismic signal processing. Methods such as support vector machines (SVMs) [17], random forests (RFs) [18], and convolutional neural networks (CNNs) [19] have been applied to seismic-noise classification tasks. More sophisticated deep learning models with diverse architectures have been developed as research has progressed. For example, Zhu et al. [20] and Zhao et al. [21] achieved high-precision P-wave picking in natural earthquakes by improving the U-Net architecture. Mousavi et al. [22] proposed the EqT model, based on an attention mechanism, which performs multiple tasks simultaneously, including event detection and P- and S-wave picking. Xiao et al. [23] proposed a dual-input S-EqT model for reliable picking in low-signal-to-noise-ratio (SNR) environments. Jiao et al. [24] proposed a microseismic P-wave arrival time-picking method that combines a gated recurrent unit with a self-attention mechanism. Yuan et al. [25] proposed a fast automated classification technique for seismic waveforms based on SegNet. Wang et al. [26] introduced an enhanced U-Net model to attain high-precision and resilient P-wave arrival detection. Subsequently, Chen et al. [27] presented a TransUNet model that merges a transformer encoder with a U-Net decoder. This model considerably improved the accuracy and resilience of microseismic P-wave and S-wave arrival detection by integrating a global attention mechanism with a local detail-preserving structure. However, most existing deep learning models are trained on data from natural earthquake networks, which differ fundamentally from coal mine microseismic scenarios. For instance, microseismic events in coal mines exhibit faster energy decay, with an effective signal duration generally shorter than 0.5 s. Additionally, the sampling frequency of coal-mine microseismic monitoring systems is typically higher than that of seismic networks. The noisy underground environment also leads to a lower SNR for microseismic signals. These differences make it difficult to directly apply pre-trained natural earthquake models to microseismic scenarios [28]. More importantly, there is a severe lack of high-quality, P-wave-labeled microseismic datasets available for model training, which fundamentally constrains the generalization ability and practical applicability of specialized models.

This paper proposes a novel method for automatically identifying microseismic P-wave arrival times, which combines physical constraints with deep learning. Firstly, we trained a U-Net-based automatic picking model using a high-precision dataset with manually labeled P-wave arrival times. We then validated its reliability by comparing it with the AR picker algorithm and the time-window energy feature method. We also conducted blind tests on data from different mining areas to evaluate the model’s generalization capabilities. Finally, we used an improved STA/LTA method and a channel spatiotemporal correlation grouping compression noise interference space to extract continuous candidate event segments from continuous waveforms. These segments were then fed into our trained arrival-time-picking model. The model’s probability distribution output mechanism converts ambiguous arrival times into quantifiable probability intervals, enabling the automatic identification of P-waves and the picking of their arrival times at coal mine sites. This methodology provides a technical approach to advance microseismic monitoring toward intelligent and practical engineering applications.

## 2. Methods

### 2.1. Model Structure

To achieve accurate and robust P-wave arrival picking in the complex and heavily noisy environment of a coal mine, we propose an automated P-wave arrival-picking model that integrates signal preprocessing with deep learning. As shown in Figure 1, the model consists of three sequentially connected core modules.

(1)Microseismic Event Detection

This module first filters real-time microseismic data to eliminate noise interference, such as power-line frequency and high-frequency noise. Subsequently, it employs the STA/LTA algorithm, combined with channel correlation analysis, for event detection, aiming to precisely identify and segment valid microseismic event waveforms from the continuous data stream.

(2)P-wave First Arrival Probability

Based on the screened valid waveform data, we constructed the training set, validation set, and test set. This module utilizes the U-Net deep learning architecture. It is trained and validated using manually annotated P-wave arrival time data to generate the optimal model capable of outputting the P-wave first arrival probability.

(3)P-wave Arrival Time Picking

This module utilizes the optimal U-Net model selected from the probability prediction stage to perform inference on the microseismic waveform data of each channel, thereby transforming the waveforms into P-wave first arrival probabilities. Finally, by analyzing the peak location of the probability curve, the P-wave onset point for each channel is determined, completing the high-precision arrival time picking and providing accurate temporal parameters for subsequent localization.

### 2.2. Microseismic Event Determination Method

The microseismic event detection process primarily includes data filtering, STA/LTA event prediction, and channel correlation analysis. The primary objective in developing this physical constraint technique was to enhance the quality of input data for the time-domain picker model. Data filtering facilitates the categorization of legitimate signals and interference. The STA/LTA event prediction approach assesses the occurrence of legitimate vibration events from multi-channel waveforms. Channel correlation analysis subsequently discards anomalous waveforms recognized as legitimate yet not associated with the same vibration event as the overall channel waveform. The core algorithm operates as follows.

#### 2.2.1. The STA/LTA Algorithm

The core idea of the algorithm is to reflect the change in amplitude by comparing the ratio of signal feature values within a short-term average (STA) window and a long-term average (LTA) window. The LTA reflects the background noise trend, while the STA captures the rapid change in signal amplitude. When the first arrival of a seismic wave occurs, the STA increases more quickly than the LTA, causing the ratio R to rise sharply. When this ratio exceeds a pre-set threshold, that moment is identified as the event’s first arrival time. The calculation for the STA/LTA algorithm is as follows.(1)$$R\left(i\right)=\frac{\mathrm{STA}(i)}{\mathrm{LTA}(i)}=\frac{\frac{1}{ns}\sum_{j=i-ns}^{i}x\left(j\right)}{\frac{1}{nl}\sum_{j=i-nl}^{i}x\left(j\right)}$$
where *i* denotes the sampling time, ns represents the short-time window length, nl denotes the long-time window length, and x(i) is the characteristic function value of the microseismic signal at time *i*, characterizing the amplitude, the energy, or their variations in the microseismic data. The principle of the STA/LTA algorithm is shown in Figure 2.

#### 2.2.2. Channel Correlation Analysis

Microseismic sensors in underground coal mines are spread across different districts, frequently resulting in a single waveform file capturing multiple vibration events. The noisy environment, including equipment vibrations and mechanical noise, can also be erroneously picked, interfering with the true signals of coal and rock fractures. To improve the accuracy of microseismic event detection within a target area and effectively suppress interference from neighboring or distant channels, this paper proposes an adaptive channel grouping algorithm that integrates multidimensional information, including spatial location, amplitude, and arrival time. The process for eliminating abnormal waveforms is shown in Figure 3.

This method dynamically identifies a set of sensors that exhibit a collaborative response by incorporating the response amplitude of each channel and the consistency of its P-wave arrival time. Even if sensors are spatially distant, if their waveform responses are highly correlated in terms of amplitude, energy, and arrival time structure, they can be intelligently grouped into the same analysis set. This significantly enhances the spatial focusing capability for local microseismic signals while effectively identifying and suppressing substantial interference from non-target areas. This strategy not only improves the data’s signal-to-noise ratio (SNR) but also reduces the potential for mutual interference between channel waveforms from different underground mining zones. It ensures, at the data source level, that the vibration waves collected within a channel originate from the same seismic event. This enhances the reliability of the data input for high-precision, automatic P-wave picking in complex, multi-channel monitoring scenarios.

### 2.3. U-Net Model

The U-Net model, based on a convolutional neural network, was first applied to biomedical image segmentation tasks [29]. Its core architecture consists of a symmetrical encoder–decoder structure with skip connections, which enables multi-scale feature fusion while preserving spatial details. Similar to an image segmentation task, P-wave picking can be precisely modeled as a time-domain signal binary classification problem in which each sample point must be classified as either a P-wave signal or background noise. Unlike traditional classification models that output a discrete arrival time label, U-Net outputs a P-wave arrival probability sequence that is time-aligned with the input data. The peak of this sequence, when it exceeds a set threshold, is identified as the arrival time, thereby achieving P-wave first-arrival picking in a coal-mine microseismic monitoring scenario. The model’s network architecture consists of a five-level encoder and a four-level decoder. The encoder progressively extracts and compresses the feature representation of the input waveform. The decoder is responsible for reconstructing the feature representation during the upsampling process. Each decoder level consists of a transposed convolution layer and a resampling convolution layer, with the number of channels gradually decreasing as the depth of the network increases. All of these convolutional and transposed convolutional layers use the ReLU activation function [30]. Ultimately, following Softmax normalization, the network produces the probability distribution for each time point associated with the P-wave and noise categories. The Softmax function is expressed as follows: (2)$$q_{m}\left(x\right)=\frac{e^{Z_{m}\left(x\right)}}{\sum_{l=1}^{2}e^{Z_{l}\left(x\right)}}$$

In the equation, m=1,2 represents the noise and P-wave types, respectively. The term Z(x) is the unnormalized value (or logit) for the input, and q(x) is the probability of the x-th sample point being a P-wave or noise. The loss function used in this paper is the cross-entropy loss function, which measures the difference between the model’s predicted probability distribution and the actual probability distribution. Its formula is as follows:(3)$$L=-\sum_{m=1}^{2}\sum_{t=1}^{n}d_{m,t}\cdot \log\left(q_{m,t}\right)$$
where *t* is the sample point index, *d* represents the true probability distribution, and *q* represents the model’s predicted probability distribution.

### 2.4. P-Wave Arrival Detection

The P-wave arrival time-picking process, based on the U-Net model proposed in this paper, is illustrated in Figure 1. The specific process is described below.

(1)Data Preprocessing and Event Detection

After the raw microseismic signals are obtained from the monitoring system, the data are first preprocessed. This includes demeaning, band-pass filtering, and normalization to remove baseline drift and suppress high-frequency and low-frequency noise. Subsequently, an event detection process is employed, which combines the Short-Term-Long-Term Average (STA/LTA) method with channel spatiotemporal correlation grouping, to identify potential microseismic events and extract corresponding candidate event segments. P-wave arrival times are then manually labeled to create the dataset used for model training.

(2)Model Training

The prepared dataset is divided into training, validation, and test sets. The model’s input is a preprocessed waveform segment, and its output is a sequence of P-wave arrival probabilities. During training, the model learns the parameters using the training set. Its performance is monitored on the validation set to prevent overfitting. The network weights are iteratively optimized to obtain the best model for P-wave first arrival picking.

(3)Model Application

The trained model is quantitatively evaluated on an independent test set. This step verifies the model’s picking accuracy, effectiveness, and robustness in real-world application scenarios.

## 3. Data and Model Training

The microseismic data utilized in this study were acquired from a monitoring system deployed within the production environment of the Zhaozhuang Coal Mine in Jincheng, Shanxi Province (as shown in Figure 4). This system comprises six monitoring substations and 48 sensors (6 × 8) deployed across three main longwall working faces, with an approximate sensor spacing of 100 m. Operating at a sampling frequency of 1000 Hz, the system enables the continuous and high-precision acquisition of microseismic signals from the mining area, providing a reliable foundation for field data in model training, validation, and testing.

The recorded data consists solely of single-component (vertical component) waveform data, with each data point spanning a duration of 2 s. To construct a high-quality, high-confidence training dataset, we adopted an “automatic pre-labeling followed by manual refinement” annotation strategy. This process included correcting P-wave arrival time annotation deviations, as well as identifying and labeling noise samples originating from instrument interference, blasting operations, or environmental noise. This meticulous procedure ensured the accuracy and representativeness of the dataset. The final constructed dataset contains 12,220 microseismic waveforms and noise records. Following data augmentation, the training set was expanded to a total of 61,100 samples. Figure 5 illustrates the signal-to-noise ratio (SNR) distribution histogram and the corresponding box plot for the partitioned dataset. Calculated following the methodology of Wang et al. [31], the SNR values range from 4.30 to 74.31, with a consistent data distribution maintained across all subsets.

### 3.1. Data Processing

#### 3.1.1. Preprocessing

Before being fed into the model, all data undergoes a standardized preprocessing workflow, which includes three key steps: demeaning, filtering, and amplitude normalization. First, the demeaning operation removes the DC component and any systematic offset from the signals, ensuring that the data is centered at zero in the time domain. Second, suitable filtering parameters are applied based on the on-site noise environment and sampling parameters to suppress low-frequency drift and high-frequency noise interference. Finally, amplitude normalization is performed to linearly scale all waveform data to the range of [−1, 1], which prevents amplitude-scale differences or outliers from affecting the model.

#### 3.1.2. Data Augmentation

To enhance the model’s generalization ability in handling signal intensity fluctuations and event timing uncertainties within a complex coal mine noise environment, we applied data augmentation techniques to the training set. These included window sliding, random amplitude scaling, and controlled Gaussian noise injection.

Specifically, the window-sliding algorithm randomly crops fixed-length waveform segments, ensuring that the P-wave first arrival point always remains within the analysis window. This reduces the model’s dependency on the precise temporal position of the event. Amplitude scaling dynamically adjusts the waveform amplitude by introducing a random scaling factor, thereby enhancing the model’s adaptability to microseismic signals of varying energy levels. The scaling factor range is 0.9 to 1.1. Gaussian noise injection superimposes noise with a mean of zero and random coefficients ranging from 0 to 0.5 times the standard deviation of the original waveform onto the source signal, thereby simulating background noise interference. Following data augmentation, the training set expands to six times its original size, with each original waveform yielding five augmented samples. These augmentation strategies work together to improve the model’s robustness, stability, and on-site applicability.

#### 3.1.3. Label Construction

For label construction, the sample labels are in the same probability distribution format as the model’s output. For pure noise samples, the P-wave class probability is uniformly set to 0, and the noise class probability is set to 1. For samples containing P-waves, triangular soft labels are constructed, centered at the manually labeled P-wave first arrival time μ. This distribution covers the time interval [μ − 0.02 s, μ + 0.02 s], peaking at μ with a probability value of 1, and linearly decaying to 0 on both sides. Using a probabilistic approach to represent the arrival time reduces the uncertainty caused by manual labeling errors and helps accelerate model convergence. A sample label example is shown in Figure 6.

### 3.2. Model Training and Validation

In this study, before model training, the original dataset was divided into a training set, a validation set, and a test set at an 8:1:1 ratio. This ensured that the data distribution was consistent and non-overlapping across all three subsets. The training set was used to learn the model’s parameters, the validation set was for hyperparameter tuning and monitoring the training process, and the test set was used for the final evaluation of the model’s performance.

The model was constructed using the PyTorch 2.5.1 deep learning framework, and the Adam optimizer was employed for parameter updates. The initial learning rate was set to $1\times 10^{-4}$ to strike a balance between convergence speed and training stability. During training, mini-batch gradient descent was utilized with a batch size of 32, optimizing both the accuracy of gradient estimation and GPU memory efficiency. To enhance the data-loading speed, a custom data loader was designed. This loader, combined with multi-threading techniques, parallelizes data reading and model computation, effectively reducing I/O waiting time and ensuring high training efficiency. Furthermore, to prevent overfitting in the later stages of training, an early stopping mechanism was implemented. This mechanism evaluates the model’s F1-score on the validation set after every 1000 iterations. If the validation F1-score fails to improve for five consecutive evaluations, the training is automatically terminated, and the model weights that achieved the highest F1-score on the validation set are restored.

### 3.3. Model Performance Evaluation

To evaluate the effectiveness of microseismic P-wave arrival time picking, we selected the mean and standard deviation of the P-wave arrival time error, along with three standard machine learning performance metrics: precision (*P*), recall (*R*), and the F1-score [32]. Their calculation formulas are as follows.(4)$$P=\frac{TP}{TP+FP}$$(5)$$R=\frac{TP}{TP+FN}$$(6)$$F1-score=2\times \frac{P\times R}{P+R}$$(7)$$\mu =\frac{1}{N}\sum_{i=1}^{N}e_{i}$$(8)$$\delta =\sqrt{\frac{1}{N}\sum_{i=1}^{N}{\left(e_{i}-\mu \right)}^{2}}$$

In these equations, TP represents the number of samples where the model’s picked arrival time is correct. FP is the number of noise instances incorrectly identified as valid arrival times. FN is the number of actual arrival times the model failed to detect. A pick is considered correct if the difference between the model’s predicted P-wave arrival time and the manually labeled actual arrival time is less than 0.02 s. *P* indicates the exactness of the model’s P-wave picking. *R* measures the completeness of the model’s P-wave picking. The F1-score is the harmonic mean of precision and recall. For the correctly identified samples (TP), we filter for those with an absolute error of less than 0.1 s. For this subset, we calculate the mean (μ, in ms) and standard deviation (δ, in ms) of the P-wave arrival time errors. The mean (μ) reflects the overall bias of the model’s predictions, while the standard deviation (δ) indicates the stability of the picks by showing the spread of the prediction results.

## 4. Results

### 4.1. Model Performance

As shown in Figure 7, the trends of the evaluation metrics during the model training process demonstrate good convergence and generalization performance. The training loss decreased rapidly in the initial phase and stabilized after approximately 6000 epochs, exhibiting no significant oscillations or increases, which indicates that the model parameters had mainly converged.

Concurrently, the precision and recall on the validation set steadily improved from approximately round 1000. Specifically, precision increased significantly from 80% to 93%, while recall also rose from 77% to 91%. The simultaneous optimization of both metrics suggests that the model exhibits strong generalization capabilities and is well suited to practical engineering applications.

### 4.2. Comparison with Traditional Methods

#### 4.2.1. Traditional Methods

To demonstrate the effectiveness of the proposed method, we conducted a comparative analysis on the test set against two well-known arrival time-picking algorithms: the AR-Picker algorithm and the Time-Window Energy Feature Method.

The AR-Picker algorithm is an automated P-wave arrival time-picking method that combines the Autoregressive model with the Akaike Information Criterion (AR-AIC) and the Short-Term/Long-Term Average Ratio (STA/LTA) [33]. Its primary advantage lies in its multi-stage, progressive signal analysis strategy, which gradually focuses on signal transition regions. This approach effectively overcomes the issues of inaccurate and unstable picking that single-method approaches face in low-SNR or complex noise environments. The method involves a straightforward workflow, a well-defined physical mechanism, and a high degree of automation. Its strong adaptability to various geological and noise conditions has led to its widespread use in the efficient processing of large-scale seismic and microseismic data. In this study, we used the ar_pick function from the ObsPy open-source seismology toolkit [34], with parameters optimized for the spectral and noise characteristics of our coal mine microseismic signals, to automatically pick the P-wave first arrival times in our dataset.

The Time-Window Energy Feature Method (hereinafter referred to as TWEFM) is a fundamental technique for the automated detection of seismic first arrival times [14]. Due to its clear physical interpretation, high computational efficiency, and strong practical applicability, it is widely used for P-wave picking in microseismic monitoring. As shown in Figure 8, the core principle of this method is as follows: The P-wave is the first body wave to arrive in a seismic sequence, and its arrival is typically accompanied by a sharp, step-like increase in local energy over a very short time. This is observed as a discontinuous jump in signal energy in the time domain. By using a sliding time window to calculate local energy and detect its inflection points, the P-wave arrival time can be effectively captured.

#### 4.2.2. Model Performance Comparison

As shown in Table 1, the model proposed in this paper outperforms the AR-Picker algorithm and the Time-Window Energy Feature Method across several key performance metrics. On the test set, our model achieved a precision of 96.60%, a recall of 90.59%, and an F1-score of 93.50%. This represents a significant improvement over both comparative methods, demonstrating its superior noise suppression and discriminative feature extraction capabilities in complex coal mine microseismic environments. This effectively reduces the risk of false positives and missed detections. In terms of arrival time accuracy, our model’s average error was only 5.49 ms, representing a more than 50% reduction compared to the AR-Picker (17.01 ms) and the Time-Window Energy Feature Method (12.64 ms). This indicates a much higher picking accuracy. Furthermore, its error standard deviation was 8.70 ms, which is significantly lower than that of the other methods, indicating that the model’s outputs are more consistent and stable.

Figure 9 illustrates the P-wave arrival time-picking performance of the model on test samples with varying amplitudes. The results show that the model’s output probability curve presents a clear, sharp, single-peak structure near the actual P-wave first arrival time, regardless of the signal’s amplitude. The peak position aligns precisely with the manually labeled arrival time, fully verifying the model’s ability to accurately capture the P-wave features and precisely localize them in time. Additionally, the probability curve remains stable at a low amplitude during non-event segments. It quickly rises during the event trigger, demonstrating intense signal-to-noise discrimination and dynamic response characteristics.

Figure 10 show the residual distribution histograms for the predicted versus manually labeled arrival times for the two traditional methods and our U-Net model on the test set. These histograms visually illustrate the differences in temporal accuracy and stability among the various methods. The residual is defined as the manually picked P-wave arrival time subtracted from the algorithm’s automatically picked time. The sign of the residual indicates whether the prediction is early or late relative to the actual value: a positive value signifies that the model’s prediction is earlier than the manual annotation (i.e., picked too early), while a negative value indicates that the prediction occurs later than the actual arrival time (i.e., picked too late). By analyzing the residuals’ central tendency, dispersion, and distribution morphology, the performance differences among the methods in terms of temporal accuracy and stability can be visually revealed. Our method’s residual distribution is highly concentrated near zero, with a prominent peak and a narrow, low-dispersion range. This indicates that it can maintain stable and accurate arrival time picking even when dealing with diverse microseismic waveforms and noise interference. In contrast, the residuals from the Time-Window Energy Feature Method and the AR-Picker algorithm are more dispersed, showing less stable predictions. The AR-Picker algorithm also exhibits a systematic negative bias, meaning its predicted arrival times are consistently later than the manually labeled reference values. This phenomenon is consistent with the findings of Li et al. [35] and may be due to the delayed response of its STA/LTA stage to energy accumulation.

#### 4.2.3. F1-Score Comparison Under Different Signal-to-Noise Ratios

As shown in Figure 11, the trend in the F1-scores for all three P-wave-picking methods at different signal-to-noise ratios (SNRs) indicates that improved signal quality generally enhances first arrival identification performance, as the F1-score for all methods increases with a rising SNR.

However, the U-Net model used in this study demonstrates a significant advantage across the entire signal-to-noise ratio (SNR) range. At low SNRs (below 20 dB), its F1-score is noticeably higher than those of both the AR-Picker and the Time-Window Energy Feature Method. The latter exhibits the poorest performance, as its reliance on thresholds or energy jumps makes it highly susceptible to noise. Conversely, at high SNRs (above 20 dB), the U-Net model consistently maintains its performance lead, while the AR-Picker’s limited noise resistance results in the lowest F1-scores. Notably, the U-Net model maintains high identification accuracy even under challenging low-SNR conditions. Its performance curve shows a gradual and smooth increase without sharp fluctuations, reflecting the model’s superior robustness and adaptability to varying noise levels.

#### 4.2.4. Computational Speed Comparison

In routine coal mine microseismic monitoring, the real-time processing capability of an algorithm is a crucial indicator of its practical utility. This is especially true when continuously processing high-sampling-rate data streams from multiple channels, as computational efficiency directly impacts the system’s response time and deployment feasibility.

To objectively evaluate the computational performance of our model, we conducted a horizontal comparison of the average time required for each method—ours, the AR-Picker algorithm, and the Time-Window Energy Feature Method—to process a single-channel, two-second waveform. This was performed on a unified hardware platform (Manufactured by Lenovo (Beijing) Co., Ltd., Beijing, China; CPU: Intel i5-1137G7 @ 2.40 GHz; GPU: NVIDIA MX450; RAM: 16 GB) using the 1222 waveform test set mentioned earlier. This comparison quantifies the time cost of each method in a real-world deployment environment. The results are shown in Table 2.

The AR-Picker algorithm (with LTA = 100 ms, STA = 10 ms and a threshold of 5, and no S-wave picking) requires a point-by-point calculation of the AIC function, leading to an average processing time of 2.97 ms for a single two-second waveform. The Time-Window Energy Feature Method, which utilizes a 100-ms sliding window and a two-point step size, relies on the energy ratio between consecutive windows to detect the P-wave phase. By incorporating prefix-sum optimization, computational efficiency is significantly boosted, reducing the time to 1.98 ms. In contrast, our U-Net-based deep learning model achieves a substantial speed advantage. By leveraging batch processing (batch size = 32) during the inference phase, it processes a single record in just 0.67 ms.

While deep learning models entail a higher training cost, their extremely low inference latency makes them particularly suitable for real-time applications, such as coal mine microseismic monitoring, which require high throughput and low latency. In the future, model lightweighting techniques could be further utilized to reduce parameter size and computational overhead. This would maintain high-precision picking performance while improving the model’s deployment feasibility on edge computing devices or resource-constrained industrial control terminals, thus pushing microseismic intelligent monitoring systems toward a more distributed, low-power, and real-time future.

## 5. Discussion

### 5.1. Model Performance in Continuous Event Sampling

To validate the engineering applicability of the model in real-world coal mine microseismic monitoring scenarios, we constructed a synthetic composite waveform sequence simulating multiple continuous events. This sequence was created by concatenating six microseismic waveforms from different sources in chronological order. The P-wave first arrivals of adjacent events were spaced at a uniform interval of approximately 2 s, which simulates a typical scenario of dense microseismic activity during production. The concatenated continuous signal was then put through our standard preprocessing workflow and fed into the trained model for automatic P-wave arrival time picking. This test was designed to evaluate the model’s ability to identify events under complex conditions, including a continuous stream of multiple events, overlapping energy, and close temporal proximity.

As shown in Figure 12, the model produced clear, sharp probability peaks at the arrival time of each microseismic event. The peak times aligned precisely with the manually labeled, accurate first arrival times. These results demonstrate that our method can not only accurately identify isolated events but also efficiently separate and precisely locate multiple P-waves within a continuous waveform. This shows its excellent temporal resolution and capability for handling multiple concurrent events.

### 5.2. Model Performance Across Different Mining Areas

To systematically evaluate the model’s generalization ability on data from mining areas that were not included in the training, we selected microseismic waveform samples from three coal mines with geological conditions and data sources independent of our primary dataset. We strictly followed the same preprocessing procedures as those used for the training data before feeding the data into the model for arrival time prediction and subsequent performance evaluation. The results are shown in Table 3.

Our model performed excellently at the Longwangou and Zhengzhuang coal mines, achieving F1-scores of 94.47% and 98.67%, respectively. This superior performance is attributed to two factors: the data sampling frequency (1000 Hz) was consistent with our training set, and the overall signal-to-noise ratio was high. This demonstrates that, with matching data acquisition parameters and good signal quality, the model achieves excellent cross-regional transferability and practical applicability.

However, the model’s performance dropped significantly during testing on data from the Changping coal mine, for which the F1-score was only 23.12%. Our analysis revealed two key reasons for this: the data from this mine generally exhibited a low signal-to-noise ratio, and the sampling frequency was 2000 Hz. The low-SNR environment increased the confusion between valid signals and background noise, resulting in numerous missed and false detections. The mismatched sampling rate likely reduced the model’s ability to capture the fine-grained temporal features of the P-wave.

To improve the model’s adaptability to heterogeneous sampling rates and complex noise environments, future work can focus on two areas. Firstly, we could utilize data augmentation strategies, such as interpolation or resampling, to create a mixed training set with multiple sampling rates. This would enhance the model’s compatibility and robustness with inputs of different time resolutions. Then, we could utilize transfer learning or domain adaptation techniques to fine-tune the pre-trained model using a small number of labeled samples from the target mining area, thereby enhancing the model’s performance in real-world deployment.

### 5.3. Model Performance Across Different Signal-to-Noise Ratio Data

To further evaluate the model’s robustness under different noise levels, we summarized the distribution characteristics of our model’s P-wave arrival time errors as a function of the signal-to-noise ratio (SNR), as shown in Figure 13. The results indicate a significant negative correlation between the dispersion of the picking error and the SNR. When the SNR is below 25 dB, the error distribution range is wide due to background noise, and some samples show significant deviations. However, when the SNR increases to 25 dB or higher, the errors quickly converge and remain stable within a ±20 ms range, demonstrating excellent noise resistance and predictive consistency.

To further quantify the model’s robustness at different noise levels, we conducted a statistical analysis of the distribution of P-wave arrival time residuals with respect to the SNR, as shown in the box plot in Figure 13. When the SNR is between 5 and 15 dB, the residual median is −18.38 ms (IQR: [−28.53, −8.22] ms), with a wide range between [−38.68, +1.93] ms. In the 15–25 dB range, the residual median significantly shrinks to −2.90 ms (IQR: [−8.70, +0.97] ms). The model’s performance becomes even more stable when the SNR is 25 dB or higher. In the 25–35 dB range, the residual median further decreased (−1.93 ms). In the 35–45 dB and 45–55 dB ranges, the residual medians are 0.00 ms and −1.93 ms, respectively, with IQRs of [−2.90, +2.90] ms and [−3.87, +0.97] ms. For even higher SNRs (>55 dB), although the sample size is small, all data points remain within their whiskers, showing the model’s high consistency and accuracy in high-SNR environments. This pattern confirms that the model is highly accurate in high-SNR environments and also suggests that high-precision arrival time picking requires a high-quality waveform acquisition system.

The reason for this lies in the model’s data preprocessing and event detection stages. The data filtering algorithms were unable to identify the spectral characteristics of the noise, thereby limiting their effectiveness. The selected STA/LTA method is also susceptible to the signal-to-noise ratio (SNR). When it is difficult to set parameters such as filter settings, short and long sliding window lengths, and thresholds, noise peaks can easily be misidentified as arrivals, leading to false positives in event detection. The accumulation of these errors across the multi-stage processing chain affects the final picking accuracy. Ultimately, the SNR is the primary limiting factor for identifying microseismic arrival times. Therefore, key research directions for improving accuracy include methods to enhance SNR, such as improved sensor deployment, channel stacking, and intelligent noise reduction, as well as the development of new noise-resistant algorithms, including multi-channel joint picking, transfer learning, and uncertainty quantification.

## 6. Conclusions

To enhance the accuracy of P-wave arrival picking for microseismic event localization in coal mines, this study designed and trained an automated microseismic P-wave arrival picking method that integrates physical constraints with deep learning. We constructed a high-quality, manually annotated microseismic waveform dataset using actual microseismic monitoring data from the Zhao Zhuang Coal Mine. The comprehensive performance of the proposed model was then systematically evaluated based on accuracy, recall, the F1-score, and the arrival time error. The experimental results demonstrate that this model significantly outperforms traditional time-window energy feature methods and the AR-Picker algorithm, achieving a mean arrival time picking error as low as 6.34 ms. This validates the model’s capability to perform high-precision automatic P-wave arrival time identification in complex coal mine environments.

To test the model’s cross-regional generalization ability, we conducted a blind validation using real data from multiple independent mining areas. The model’s transfer performance was excellent in the Longwang Gou and Zhengzhuang coal mines, which share consistent sampling frequencies (1000 Hz) and high signal-to-noise ratios, achieving F1-scores of 94.47% and 98.67%, respectively. This demonstrates good engineering adaptability. However, the model’s F1-score dropped significantly to 23.12% in the Changping coal mine, which features a higher sampling frequency (2000 Hz) and more substantial noise interference. This drop indicates the model’s sensitivity to changes in acquisition parameters, highlighting the necessity of considering the dataset’s sampling parameters during application.

This study validates the superior performance of neural networks in microseismic signal processing, representing a significant practical application of deep learning technology within the field of intelligent disaster monitoring for coal mines. Future work will be pursued in two primary directions. On the one hand, we will introduce strategies such as data augmentation, transfer learning, and domain adaptation to further enhance the model’s pick-up capabilities across diverse sampling conditions and complex noise environments. The objective is to facilitate the engineering deployment and real-time application of neural network technology across coal mine sites. On the other hand, we will actively explore automatic tuning mechanisms for the intrinsic trade-off between model accuracy, noise robustness, and inference efficiency. The objective is to achieve a synergistic improvement in both performance and efficiency, thereby providing a more reliable and effective intelligent monitoring solution for practical industrial scenarios.

## Figures and Tables

**Figure 1 sensors-25-07103-f001:**
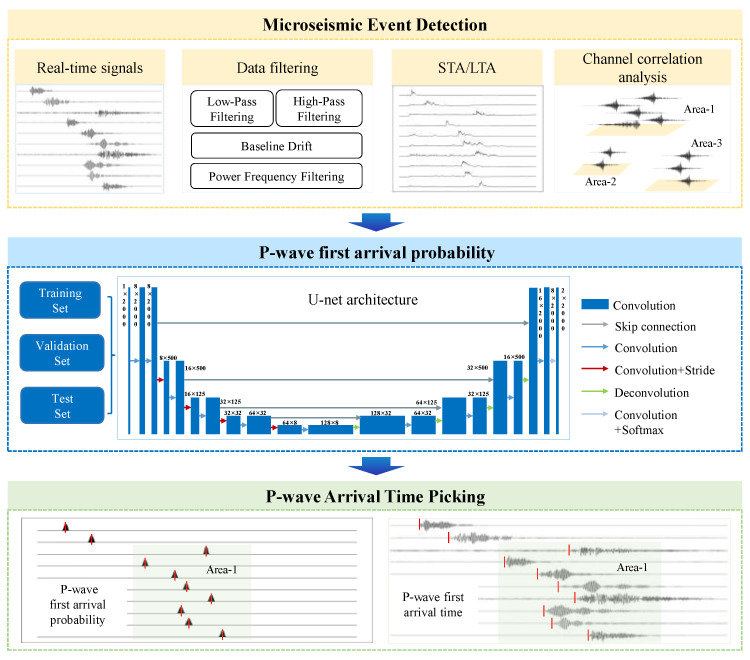
Automatically capture model structure upon arrival of microseismic P-wave.

**Figure 2 sensors-25-07103-f002:**
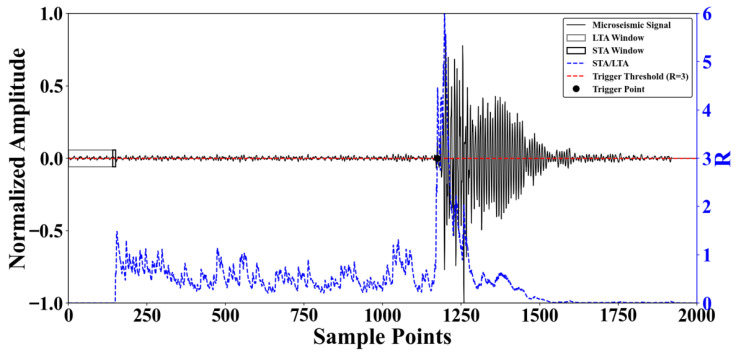
Principle of STA/LTA algorithm.

**Figure 3 sensors-25-07103-f003:**
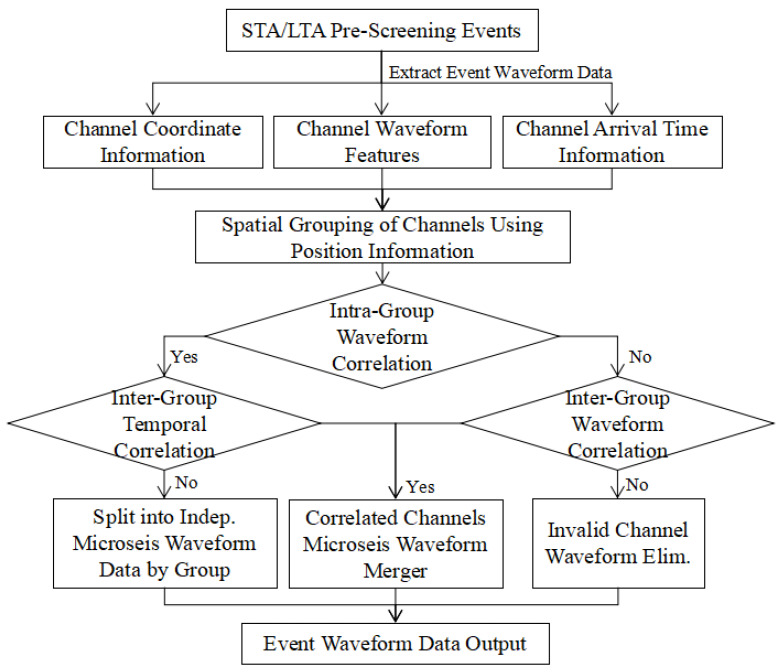
Chanel correction analysis process.

**Figure 4 sensors-25-07103-f004:**
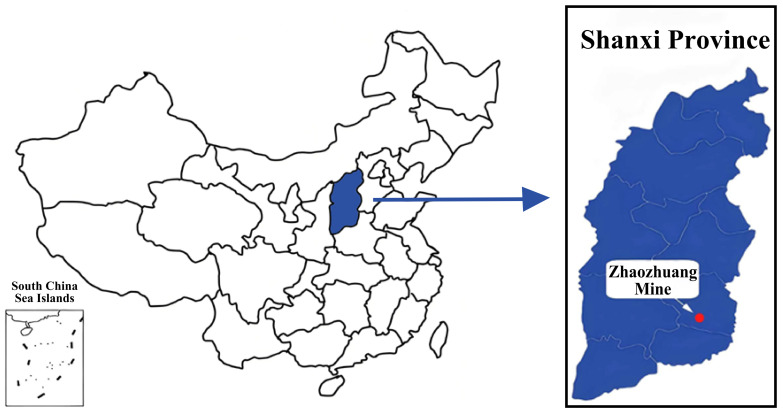
The territory of the Zhaozhuang Mine.

**Figure 5 sensors-25-07103-f005:**
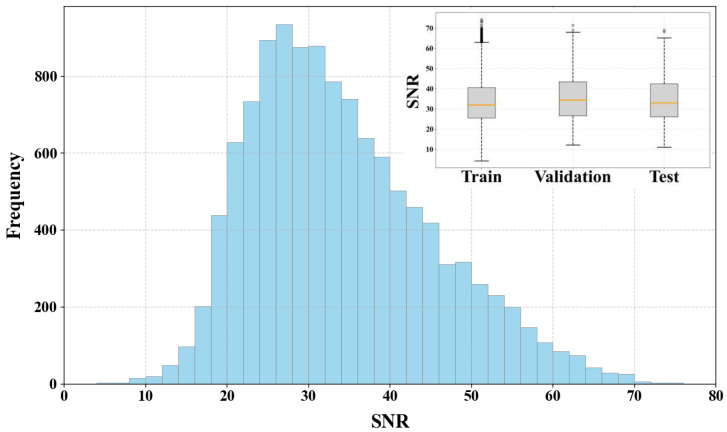
SNR distribution.

**Figure 6 sensors-25-07103-f006:**
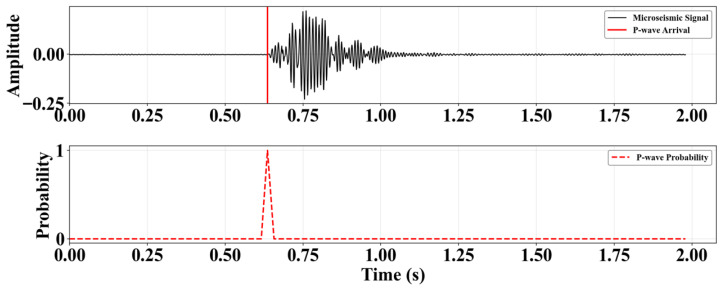
Sample label example.

**Figure 7 sensors-25-07103-f007:**
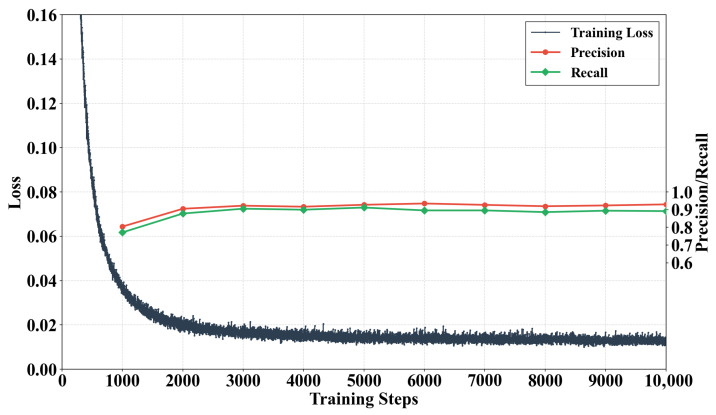
Performance evaluation of the model on the validation set.

**Figure 8 sensors-25-07103-f008:**
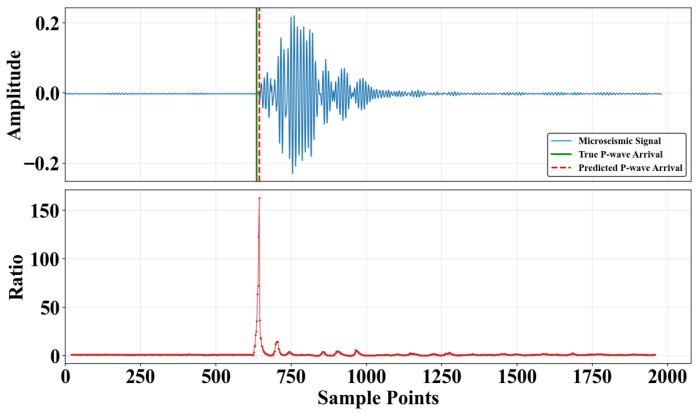
Example of time-window energy feature prediction.

**Figure 9 sensors-25-07103-f009:**
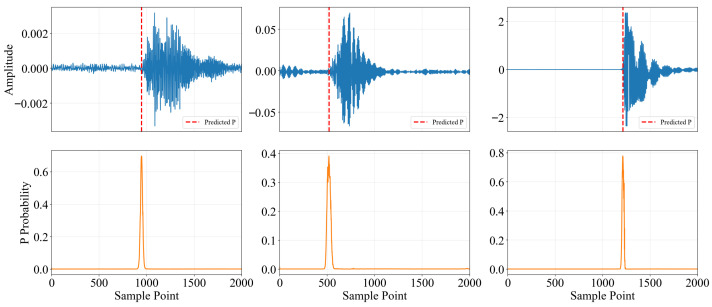
Results of picking up different energy events at the designated time.

**Figure 10 sensors-25-07103-f010:**
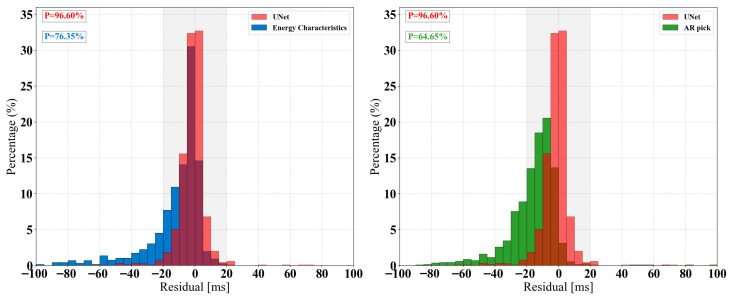
Comparison between U-Net and the other two methods.

**Figure 11 sensors-25-07103-f011:**
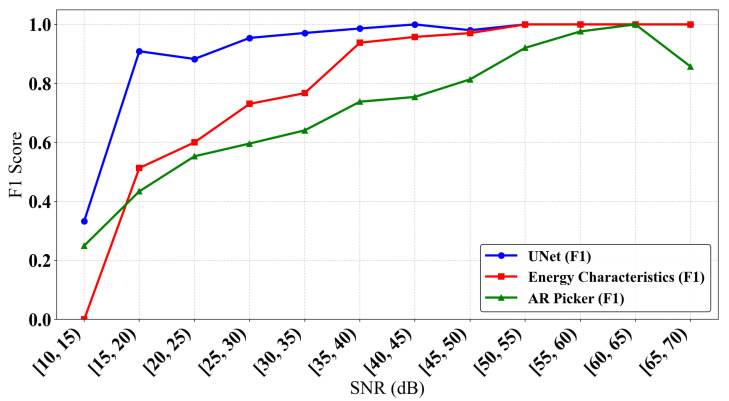
Comparison of model F1 scores at different signal-to-noise ratios.

**Figure 12 sensors-25-07103-f012:**
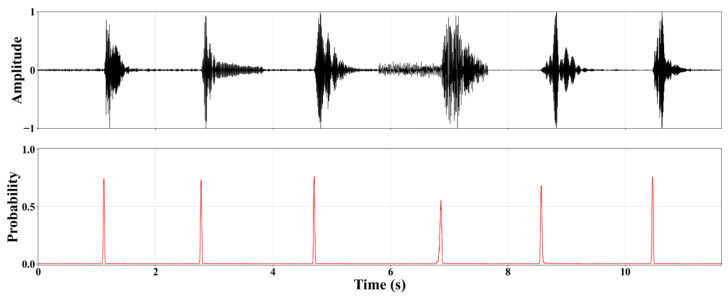
Continuous vibration data acquisition timing.

**Figure 13 sensors-25-07103-f013:**
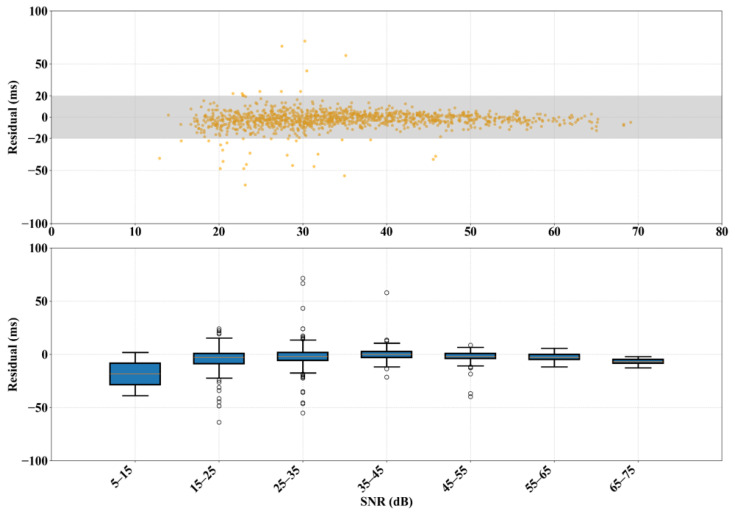
Distribution of P-wave arrival time error versus signal-to-noise ratio.

**Table 1 sensors-25-07103-t001:** Three arrival-time pick-up model comparison results parameters.

Parameter	AR Pick	TWEFM	Proposed Model
**Precision (%)**	64.65	76.35	96.60
**Recall (%)**	64.65	76.35	90.59
**F1-score (%)**	64.65	76.35	93.50
**Mean (ms)**	17.0103	12.6440	5.4871
**Standard Deviation (ms)**	15.6892	17.3362	8.6970

**Table 2 sensors-25-07103-t002:** Comparison of processing speeds for different algorithms.

Parameter	AR Pick	TWEFM	Proposed Model
**Min. Processing Time (ms)**	1.67	1.19	—
**Max. Processing Time (ms)**	15.97	4.16	—
**Avg. Processing Time (ms)**	2.97	1.98	0.67
**Method Characteristics**	traversing AIC function	Energy ratio calculation	Multi-channel parallel computation
**Efficiency Evaluation**	Lower	Higher	High

**Table 3 sensors-25-07103-t003:** Model performance in other mining areas.

Parameter	Longwang Gou	Zhengzhuang	Changping
**Number of Events**	100	75	159
**Sampling Frequency**	1000	1000	2000
**SNR Range**	18.08∼61.48	24.14∼63.71	9.25∼49.64
**F1-score (%)**	94.47	98.67	23.21

## Data Availability

Data are contained within the article.

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
