# Peer review of "High-Precision Coal Mine Microseismic P-Wave Arrival Picking via Physics-Constrained Deep Learning"

_sensors, 2025, doi:10.3390/s25237103_

Round 1
Reviewer 1 Report
Comments and Suggestions for Authors
The manuscript addresses a relevant and practical topic in automatic P-wave arrival picking for coal mine microseismic monitoring and presents a technically competent application of deep learning. The paper is well organized, and the results appear quantitatively consistent. However, the overall originality is limited, and the technical contribution beyond existing models remains unclear. The proposed framework is largely based on a U-Net architecture, which has already been widely adopted in P-wave and seismic phase picking studies (e.g., PhaseNet, Earthquake Transformer, S-EqT).
The work has potential value as an engineering application study, but a major revision is needed before it can be considered for publication.
1. The manuscript does not provide adequate details about the input microseismic signals used for model training and validation. For reproducibility and transparency, the authors should clearly describe the dataset characteristics, including waveform properties, sampling parameters, SNR distribution, and representative signal examples.
2. A more thorough comparison with recent studies (2021–2025) is required to clarify the methodological or theoretical contributions beyond applying an existing deep learning model to a new dataset. The authors should explicitly explain what is unique in their “physics-constrained” approach and how it advances current practices.
Author Response
Dear Reviewer,
We sincerely appreciate your comments on improving the quality of this manuscript. Based on these suggestions, we have comprehensively revised the whole paper including grammar and some small mistakes, and we believe that it is much stronger as a result of the changes.
Comments [IB1]: The manuscript does not provide adequate details about the input microseismic signals used for model training and validation. For reproducibility and transparency, the authors should clearly describe the dataset characteristics, including waveform properties, sampling parameters, SNR distribution, and representative signal examples.
Response:
We thank the reviewer for valuable suggestions. As you have pointed out, the description of the microseismic signal details used for training and validation was insufficient in the original manuscript. In the revised manuscript, we have now provided a detailed description of the relevant data in Section 3, including key information such as: the use of single-component waveforms, a sample duration of 2 seconds, the data source, and the number of samples. Furthermore, we have relocated the subsequent analysis of the dataset's Signal-to-Noise Ratio (SNR) distribution to follow this paragraph, which renders the overall narrative logic more coherent and clear. The specific modifications are highlighted in yellow on pages 7-8, lines 222-240.
Commented [IB2]: A more thorough comparison with recent studies (2021–2025) is required to clarify the methodological or theoretical contributions beyond applying an existing deep learning model to a new dataset. The authors should explicitly explain what is unique in their “physics-constrained” approach and how it advances current practices.
Response:
We thank the reviewer for valuable comments. We fully concur with the issue raised. Following a comprehensive literature review, we have supplemented the manuscript with recent advancements in U-Net-based arrival-time picking, summarizing their innovative points and comparing them with the present study. This comparison has led us to affirm that the core innovations of our study are as follows.
Unlike existing research that employs physical rules as post-processing steps or constraints within the loss function, this work integrates data filtering, an optimized STA/LTA energy ratio, and multi-channel signal correlation criteria as a pre-processing module that precedes the deep learning model's training. We propose a "channel correlation-enhanced mechanism" oriented towards underground microseismic scenarios. Based on the geometric distribution properties of the sensor array in the coal mine, we constructed a spatio-temporal correlation constraint module. This module utilizes the cross-correlation peak delay from adjacent channel signals as a priori constraint, compelling the network to maintain spatial consistency during prediction. This mechanism effectively suppresses single-channel mis-picks caused by localized disturbances (such as equipment vibrations and mechanical noise). It not only improves the data signal-to-noise ratio (SNR) but also reduces the likelihood of mutual interference between channel waveforms from different regions of the mine. This guarantees, at the data source level, that the vibration waves collected within a channel originate from the same seismic event, thereby ensuring the data more closely represents the true coal and rock fracture signals. Subsequently, the U-Net model is trained to capture key features, achieving a bi-directional synergistic optimization of "physics-constrained and data-driven" approaches, thereby enhancing the accuracy of P-wave arrival picking. In summary, the innovation of this study lies not only in the integration of physics with deep learning, but more significantly in achieving the fusion of upfront physical constraints with deep learning. This design preserves the robust feature extraction capabilities of the U-Net architecture while endowing the model with the ability to resist interference from intense noise and non-stationary background conditions encountered in downhole environments. This constitutes the fundamental breakthrough that distinguishes this work from existing research.
In the revised manuscript, we have undertaken a systematic restructuring and refinement of the methodology section in Sections 2.1 and Section 2.2. Incorporating feedback from other experts, we have optimised the data augmentation methods and corrected the model calculation results (Section 4.2) to provide further substantiation for the aforementioned innovative claims. The specific modifications are highlighted in yellow.
These modifications underscore the dual breakthroughs of this work:
(1) Theoretical level: proposing the novel paradigm of "pre-embedded physical constraints" to replace traditional a posteriori calibration;
(2) Applied level: incorporating additional data augmentation suggestions from peer reviewers to achieve a 96.6% pick accuracy rate (manually verified standard) on real-world data, thereby providing an interpretable solution for industrial scenarios.
We extend our gratitude once more to the reviewers for your insightful corrections, which have rendered the theoretical contributions of this research clearer and more robust.

Reviewer 2 Report
Comments and Suggestions for Authors
The manuscript presented a physics-constrained deep learning model for coal mine microseismic P-wave arrival picking. It will be reconsidered based on the responses to the following questions/comments.
1. In this work, probabilistic inference of the arrival time is a key outcome. However, the introduction discussed only deterministic machine learning. A concise discussion on statistical and probabilistic inference in machine learning for seismic analysis should be included in Section 1. Some representative references are:
"Bayesian synergistic metamodeling (BSM) for physical information infused data-driven metamodeling." Computer Methods in Applied Mechanics and Engineering 419 (2024): 116680.
"Probabilistic seismic inversion based on physics-guided deep mixture density network." Petroleum Science 21, no. 3 (2024): 1611-1631.
"Machine learning–based ground motion models for shallow crustal earthquakes in active tectonic regions." Earthquake Spectra 39, no. 4 (2023): 2406-2435.
2. How is the P-wave arrival probability obtained?
3. How is the probability curve designed?
4. How many sensors and monitored signals are utilized in the analysis?
5. In Section 3, how is the correction conducted? How is the noise filtered?
6. Regarding the data augmentation, how many samples are augmented? Is the augmentation valid for general situations? How is the overall performance without the augmentation?
7. What are the settings of the triangular distribution?
8. How is the generalization performance of the proposed model?
Author Response
Dear Reviewer,
We sincerely appreciate your comments on improving the quality of this manuscript. Based on these suggestions, we have comprehensively revised the whole paper including grammar and some small mistakes, and we believe that it is much stronger as a result of the changes.
Commented [IB1]: In this work, probabilistic inference of the arrival time is a key outcome. However, the introduction discussed only deterministic machine learning. A concise discussion on statistical and probabilistic inference in machine learning for seismic analysis should be included in Section 1. Some representative references are:
"Bayesian synergistic metamodeling (BSM) for physical information infused data-driven metamodeling." Computer Methods in Applied Mechanics and Engineering 419 (2024): 116680.
"Probabilistic seismic inversion based on physics-guided deep mixture density network." Petroleum Science 21, no. 3 (2024): 1611-1631.
"Machine learning-based ground motion models for shallow crustal earthquakes inactive tectonic regions." Earthquake Spectra 39, no. 4 (2023): 2406-2435.
Response:
We thank the reviewer for your suggestions. We fully concur with the issue raised. Following a comprehensive literature review, we have supplemented the manuscript with recent advancements in U-Net-based arrival-time picking, summarizing their innovative points and conducting a comparison with the present study.
In contrast to existing research that employs physical rules as post-processing steps or constraints within the loss function, the present study integrates an optimized STA/LTA energy ratio and multi-channel signal correlation criteria as a pre-processing module that precedes the deep learning model's training. We propose a "channel correlation-enhanced mechanism" oriented towards the underground microseismic scenario. Based on the geometric distribution characteristics of the sensor array in the coal mine, we constructed a spatio-temporal correlation constraint module. This module utilizes the cross-correlation peak delay from adjacent channel signals as a priori constraint, compelling the network to maintain spatial consistency during its prediction process. The specific modifications are highlighted in yellow in Section 2.1 and Section 2.2.
Commented [IB2]: How is the P-wave arrival probability obtained?
Response:
Thank you for the reviewer's question. After the model is trained and the waveform data of the input time series is input, the model will output a P-wave arrival probability and a noise probability at each time step . Among them, represents the probability of the P-wave arriving at time . The probability is generated by the softmax activation function of the last layer of the network, and the cross-entropy loss function is used for optimization during the training process. It is worth noting that the target label is not a single-point pulse, but a triangular distribution centered on the real P-wave arrival time, as shown in Figure 5, to increase a certain fault tolerance. Therefore, the "arrival probability" of the P wave is not obtained thru post-processing, but is a continuous confidence curve in the time dimension directly learned by the network. We have made a clearer description of the probability distribution in Section 2.3 and Section 3.1.
Commented [IB3]: How is the probability curve designed?
Response:
We thank the reviewer for your attention to this point. The P-wave arrival probability curve in this method is not manually designed or generated through post-processing; rather, it is a time-series confidence result output directly by the model. Specifically, the network predicts a probability for each time step of the input waveform, indicating whether it belongs to the P-wave class or the noise class. The shape of this probability curve is jointly guided by the training target (i.e., the triangular distribution centered on the true arrival time) and the loss function. Ultimately, it reflects the model's confidence in the potential arrival time of the P-wave. As illustrated in Figure 5, the P-wave arrival probability curve shown is generated by inputting the microseismic signal (shown above it) into the model. Therefore, this probability curve is a natural product of end-to-end learning, not a predefined template or a post-processing result.
Commented [IB4]: How many sensors and monitored signals are utilized in the analysis?
Response:
We thank the reviewer for your valuable inquiry. The data used in this study were sourced from 6 microseismic monitoring stations deployed underground in a coal mine in Shanxi, China. Each station is equipped with 8 single-component geophones, totaling 48 independent observation channels (6×8) In the original dataset, 12,220 valid microseismic event waveforms were labeled; an additional 48,880 noisy samples were generated via a data augmentation strategy, resulting in a total of 61,100 waveform records for model training and evaluation.
In the revised manuscript, we have comprehensively supplemented and clarified this key information in the third paragraph of the Introduction, including the data source, sensor configuration, signal type (single-component), waveform length (2 seconds), and the sample quantities at each stage. We have provided a clearer description of this in Section 3. The specific modification are highlighted in yellow on Pages7-8, line 222-229.
Commented [IB5]: In Section 3, how is the correction conducted? How is the noise filtered?
Response:
We thank the reviewer for raising this important question. We wish to clarify that the "correction" mentioned in Section 3 refers only to the label refinement during the dataset construction phase, and not to any post-processing adjustment of the model's prediction results.
Specifically, in constructing the training dataset, we employed a two-stage "automatic initial picking + manual refinement" labeling process. First, a basic picker (e.g., STA/LTA) was used for initial picking. Subsequently, experienced analysts manually reviewed and corrected any erroneously labeled P-wave arrivals. Concurrently, events contaminated by blasting, instrument malfunctions, or other non-tectonic noise were discarded to ensure both the accuracy of the labels and the representativeness of the data. The error (for model evaluation) was calculated by subtracting the predicted P-wave arrival time from the manually labeled arrival time. It must be emphasized that no additional "correction" operations were performed on the model's output during the inference phase.
Regarding noise filtering, the data filtering work in this study primarily consisted of the following steps: First, during the waveform acquisition process, low-pass, high-pass, and power-line filtering methods were used to reduce noise interference. Next, a combination of STA/LTA event pre-judgment and channel correlation analysis was employed to discard partial invalid events. Finally, prior to data pre-processing and data normalization, all waveforms underwent a band-pass filter pre-processing step to suppress low-frequency drift and high-frequency interference. These specific details have been described in the revised manuscript in yellow in Section 2.2 and Section 3.1.
Commented [IB6]: Regarding the data augmentation, how many samples are augmented? Is the augmentation valid for general situations? How is the overall performance without the augmentation?
Response:
We thank the reviewer for your valuable question regarding the data augmentation section. Inspired by your comments, we have conducted further systematic comparative experiments and have optimized our data augmentation strategy.
In the revised manuscript, we now employ three data augmentation methods that have clear physical significance and are widely used in seismic and microseismic signal processing: (1)Random time shifting; (2) Adding Gaussian noise (with a noise intensity of 0 to 0.5 times the standard deviation of the original signal); and (3) Random amplitude scaling (with scaling factors ranging from 0.9 to 1.1). These operations simulate common signal variations observed in practice, such as arrival-time uncertainty, ambient noise interference, and differences in instrument response. Therefore this method has good applicability in both natural earthquake and induced microseismicity scenarios. Based on this strategy, we generated five augmented samples for each waveform in the original training set, adding a total of 48,880 augmented data points. This resulted in a final training and evaluation dataset comprising 61,100 waveforms. After testing, this revised augmentation strategy improved the model's accuracy from 93% to 96%.
We have now added the specific augmentation factor (i.e., the number of generated samples per original waveform) in Section 3.1.2 of the revised manuscript to avoid ambiguity. The specific modifications are highlighted in yellow on pages 8-9, lines 261-266.
Commented [IB7]: What are the settings of the triangular distribution?
Response:
Thank you for the reviewer's question. In this study, the triangular probability labels span the time interval seconds (i.e., covering 20 sampling points before and after the target time, totalling 41 points). They exhibit linear decay from the peak value, decreasing to zero at the interval's endpoints. This approach mitigates uncertainty arising from manual annotation bias while accelerating model convergence. We have supplemented these details in Section 3.1.3 of the revised manuscript to provide a more comprehensive method description. The specific modifications are highlighted in yellow on page 9, lines 272-274.
Commented [IB8]: How is the generalization performance of the proposed model?
Response:
We thank the reviewer for raising this important question. In the manuscript, we evaluated the model's generalization performance based on its results on the test set. It should be noted that the primary focus of this study is the development of a model for a specific, single mine. This approach is analogous to the seismology field, where arrival-time models are often region-specific.
Furthermore, to more thoroughly validate the model's generalization capability, we conducted additional cross-mine blind tests. We tested the model on two mining areas that were not included in the training data. These two areas had the same sampling frequency and a favorable data signal-to-noise ratio (SNR). The model achieved F1-scores of 94.47% and 98.67%, respectively. These experimental results demonstrate that the model possesses good generalization capability under similar conditions.

Reviewer 3 Report
Comments and Suggestions for Authors
Determining signal onset time is a fairly common task, for example, when segmenting a voice track into phonemes or the arrival time of a neurosignal in medicine. However, a task that prioritizes signal arrival time accuracy is quite rare. The article demonstrates significant improvements in terms of speed and signal arrival time accuracy compared to algorithmic methods (Time-Window Energy Feature Method and AR-Picker). The generalization ability of the model is quite high (using the Longwang Gou and Zhengzhuang oil fields as examples), but it exhibits poor stability on highly noisy data (Changping oil field; the problem of increased frequency is not critical for convolutional models). The scientific value lies in the application
of the U-net architecture (a nearly classical version was used; classical methods should yield similar results) to the problem of determining signal arrival time (with low noise).
Comments:
1. What is the sample size? The histogram suggests that it is approximately 60,000 signals.
2. How much did the dataset grow with augmentation? A breakdown by all types of dataset expansion would be desirable. Data augmented with Gaussian noise (even though it has very low amplitude) is especially interesting.
3. What are the tensor sizes in the U-net architecture? The figure suggests that they increase by approximately 2 times with compression, but this is unlikely to be the case.
4. In the case of severe noise, the initial weak point of a signal is difficult to determine from its amplitude. Consequently, the time domain for determining such signals must be wider, otherwise, an unknown moment is searched for before the signal. As a result, the entire method seems ineffective on data with noise levels above a certain threshold.
5. In Fig. 10 shows a histogram of the distribution of signal timing predictions from the U-net architecture. It shows that the majority of predictions occur after the signal's arrival, in contrast to analytical methods, where predictions are overwhelmingly made before the signal. The question is how critical is this type of prediction for the problem and to what extent are such predictions related to the noise level? Could it be that noise shifts predictions further to the right along the time axis?
Author Response
Dear Reviewer,
We sincerely appreciate your comments on improving the quality of this manuscript. Based on these suggestions, we have comprehensively revised the whole paper including grammar and some small mistakes, and we believe that it is much stronger as a result of the changes.
Commented [IB1]: What is the sample size? The histogram suggests that it is approximately 60,000 signals.
Response:
We thank the reviewer for your question. This study utilized a total of 61,100 waveform samples, which comprises 12,220 original labeled signals and 48,880 augmented samples generated through data augmentation of the training set. We have now added detailed information regarding these dataset quantities in Section 3. Furthermore, we have relocated the Signal-to-Noise Ratio (SNR) distribution histogram (which now reflects the original dataset's distribution) to follow this paragraph, rendering the overall narrative logic more coherent and clear. The specific modification are highlighted in yellow on page 8, lines 234-240 and figure 5.
Commented [IB2]: How much did the dataset grow with augmentation? A breakdown by all types of dataset expansion would be desirable. Data augmented with Gaussian noise (even though it has very low amplitude) is especially interesting.
Response:
Inspired by your comments, we have conducted further systematic comparative experiments and have optimized our data augmentation strategy. In the revised manuscript, we now employ three data augmentation methods that have clear physical significance and are widely used in seismic and microseismic signal processing: (1) Random time shifting; (2) Adding Gaussian noise (with a noise intensity of 0 to 0.5 times the standard deviation of the original signal); and (3) Random amplitude scaling (with scaling factors ranging from 0.9 to 1.1).
These operations simulate common signal variations observed in practice, such as arrival-time uncertainty, ambient noise interference, and differences in instrument response. Therefore this method have good generalizability in both natural earthquake and induced microseismicity scenarios. Based on this strategy, we generated 5 augmented samples for each waveform in the original training set, adding a total of 48,880 augmented data points. This resulted in a final training and evaluation dataset comprising 61,100 waveforms. After testing, this revised augmentation strategy improved the model's accuracy from 93% to 96%. We have now added a detailed elaboration of the data augmentation in Section 3.1.2 of the revised manuscript. The specific modification are highlighted in yellow on pages 8-9, lines 261-266.
Commented [IB3]: What are the tensor sizes in the U-net architecture? The figure suggests that they increase by approximately 2 times with compression, but this is unlikely to be the case.
Response:
We thank the reviewer for their meticulous observation. In the U-Net variant employed in this study, a convolution with a kernel size of 7 and a stride of 4 is used. Each downsampling operation compresses the temporal dimension to one-fourth of its original size, while simultaneously doubling the number of channels (i.e., .
For an input length of 2000, as an example, the tensor shapes at each layer (Batch Size Channels Temporal Length) are as follows:
Input:
After initial convolution:
Encoder layer outputs:
Layer 0:
Layer 1:
Layer 2:
Layer 3:
Bottleneck layer:
The decoder then symmetrically upsamples (temporal length 4, channels 2) using transposed convolutions, fusing the results with the corresponding skip connections. We have now clearly annotated the channel counts and temporal dimensions in Figure 1 to prevent any misunderstanding.
Commented [IB4]: In the case of severe noise, the initial weak point of a signal is difficult to determine from its amplitude. Consequently, the time domain for determining such signals must be wider, otherwise, an unknown moment is searched for before the signal. As a result, the entire method seems ineffective on data with noise levels above a certain threshold.
Response:
We thank the reviewer for your suggestion. Data quality is a critical factor determining the precision of P-wave arrival-time picking. Although this study employed physical constraints, such as data filtering, STA/LTA event pre-judgment and channel correlation analysis, successfully improved the accuracy to 96.6% (within a 20ms error margin), which compares favorably to the 50ms error margin reported in similar studies, we acknowledge the model's current limitations.
Currently, when the SNR is below 25 dB, the model is affected by background noise, resulting in a wider error distribution. However, the focus of this study is specifically on the P-wave arrival-time picking task, and we have already clearly defined the model's minimum SNR requirements. The fine-grained filtering of data under low-SNR conditions is considered preliminary work to this study. Our team is already conducting related research on this topic, which will be discussed in future publications.
Commented [IB5]: In Fig. 10 shows a histogram of the distribution of signal timing predictions from the U-net architecture. It shows that the majority of predictions occur after the signal's arrival, in contrast to analytical methods, where predictions are overwhelmingly made before the signal. The question is how critical is this type of prediction for the problem and to what extent are such predictions related to the noise level? Could it be that noise shifts predictions further to the right along the time axis?
Response:
We are very grateful for your meticulous observation and valuable comments regarding our research. In response to the issue you identified in Figure 10, we have conducted a detailed analysis and discussion, which we wish to clarify further here.
The original Figure 10 indicated that the proportion of predictions in the 0-5 ms interval (i.e., predictions slightly earlier than the true arrival time, defined as "positive residuals" — this definition is now described in the revised manuscript, Section 4.2.2, page 13, lines 380-388.) was higher than in the -5 to 0 ms interval. We did note this phenomenon; however, because the statistical analysis of all valid prediction samples showed that the ratio of positive residuals (prediction earlier than true arrival) to negative residuals (prediction later than true arrival) was essentially balanced, at approximately 50% each, we did not discuss this in greater depth.
As you correctly pointed out, both noise levels and model training can influence this. After comprehensively considering the opinions of other reviewers, we optimized the data augmentation strategy for the training set. This resulted in a 3% improvement in the model's accuracy, and the aforementioned phenomenon was significantly mitigated. Regarding the potential influence of noise on prediction bias, it can be observed in Figure 13 that the positive and negative residuals maintain a good balance when the SNR is higher than 15 dB.
In the revised manuscript, we have updated Figure 10 to reflect the model's results following the optimized data augmentation strategy. Concurrently, we have added an explanation of how the residuals are calculated in Section 4.2.2 to make the overall narrative logic more coherent and clear. In summary, we believe the prediction bias of this U-Net model is, on the whole, manageable. Of course, we completely agree with your concern regarding the relationship between the prediction mechanism and noise. In future work, we will further investigate the enhancement of detection capabilities for coal mine vibration events under different noise levels.
Once again, we sincerely thank you for your constructive feedback, which has been crucial in enhancing the rigor and completeness of this study.

Reviewer 4 Report
Comments and Suggestions for Authors
The manuscript presents a physics-constrained deep learning framework for high-precision P-wave arrival picking in coal mine microseismic monitoring. The research topic is relevant and contributes to intelligent mining safety. However, several aspects related to dataset representativeness, model formulation, and result interpretation require further clarification before the paper can be accepted.
1. The dataset covers a limited number of mining areas, which may restrict generalisation. It is better to clarify how variations in geological conditions, sensor depth, and sampling frequency were handled. Including a cross-site validation or data distribution summary would improve the reliability of the results.
2. The paper frequently mentions the physics-constrained mechanism, but its mathematical definition remains vague. The authors should clearly describe how physical consistency is imposed during model training, such as through waveform continuity, arrival-time smoothness, or energy-based regularisation.
3. The architecture description lacks key information. Please give the number of layers, kernel sizes, activation functions, and training parameters such as learning rate, optimizer, and batch size. A schematic showing the data flow and feature dimensions would make the study easier to reproduce.
4. Although the model achieves good accuracy, statistical indicators such as confidence intervals or variance across repeated tests should be reported. Comparing the model against simpler baselines like STA/LTA or CNN-only architectures would help highlight the advantages of the proposed physics-constrained design.
5. The proposed framework involves balancing multiple goals, including accuracy, noise robustness, and computational efficiency. Future studies could draw on systematic multi-objective optimisation frameworks to address these trade-offs more effectively. In particular, “Multi-objective optimisation of surveillance camera placement for bridge–ship collision early-warning using an improved non-dominated sorting genetic algorithm” provides a useful methodological reference for structured optimisation and could inspire parameter-tuning strategies in similar physics-constrained learning tasks.
Author Response
Dear Reviewer,
We sincerely appreciate your comments on improving the quality of this manuscript. Based on these suggestions, we have comprehensively revised the whole paper including grammar and some small mistakes, and we believe that it is much stronger as a result of the changes.
Commented [IB1]: The dataset covers a limited number of mining areas, which may restrict generalisation. It is better to clarify how variations in geological conditions, sensor depth, and sampling frequency were handled. Including a cross-site validation or data distribution summary would improve the reliability of the results.
Response:
We sincerely thank you for your valuable comments. As you correctly point out, differences in mining conditions can lead to variations in model performance. The intended application of this research is for a microseismic monitoring system deployed at a specific coal mine. The methodology involves training the model on a large volume of manually labeled data from that particular mine, and subsequently using it to process P-wave picking tasks for microseismic events within that same mine. Therefore, the present study focuses primarily on the data processing, model training, and application for a single mining area.However, to further explore the influence of the microseismic system's sampling parameters on model accuracy, we also conducted cross-mine blind tests. The results indicated that the model can maintain a certain level of detection performance in unknown mining areas, demonstrating a degree of generalization capability. Nevertheless, when the sampling frequency and Signal-to-Noise Ratio (SNR) varied, the model's performance did indeed decline significantly. This reflects that the current model is highly sensitive to these parameters.
This phenomenon is highly consistent with research on the generalization of deep learning models in the field of seismology. For example, when Zhu and Beroza (2019) proposed PhaseNet, their model was trained entirely on data from the Northern California Earthquake Data Center (NCEDC) and validated with excellent performance on regional North American data. However, subsequent studies (including high-impact work) have pointed out that such models, trained in a specific region, experience a significant performance drop when transferred to regions with different geological structures. Specifically, a comparative study by Jiang et al. (2021) in Earthquake Science explicitly noted: "It has been demonstrated from the corresponding research that due to the differences in seismic waveforms found in different geographical regions, the picking performance is reduced when the two models are applied directly to the detection of the Yangbi and Maduo earthquakes. PhaseNet has a higher recall than EQTransformer, but the recall of both models is reduced by 13%-56% when compared with the results reported in the original papers." The research by Jiang et al. (Earthq Sci, 2021, 34(5): 425–435, doi:10.29382/eqs-2021-0038) strongly corroborates the "region-specific model" phenomenon (i.e., the model performs excellently in its training region, but its cross-regional generalization is limited). Given that a large volume of manually labeled P-wave data is available at the coal mine site, we are able to obtain a sufficiently large training dataset. This enables the construction of a "one mine, one model" approach for practical, on-site applications of microseismic P-wave picking.
Commented [IB2]: The paper frequently mentions the physics-constrained mechanism, but its mathematical definition remains vague. The authors should clearly describe how physical consistency is imposed during model training, such as through waveform continuity, arrival-time smoothness, or energy-based regularisation.
Response:
We thank the reviewer for your valuable comments. In contrast to existing research that employs physical rules as post-processing steps or as constraints within the loss function, this work integrates data filtering, an optimized STA/LTA energy ratio, and multi-channel signal correlation criteria as a pre-processing module that precedes the deep learning model's training.
We propose a "channel correlation-enhanced mechanism" oriented towards the underground microseismic scenario. Based on the geometric distribution characteristics of the sensor array in the coal mine, we constructed a spatio-temporal correlation constraint module. This module utilizes the cross-correlation peak delay from adjacent channel signals as a priori constraint, compelling the network to maintain spatial consistency during its prediction process. This mechanism effectively suppresses single-channel mis-picks caused by localized disturbances (such as equipment vibrations and mechanical noise). It not only improves the data signal-to-noise ratio (SNR) but also reduces the potential for mutual interference between channel waveforms from different regions of the mine. This ensures, at the data source level, that the vibration waves collected within a channel originate from the same seismic event, thereby ensuring the data more closely represents the true coal and rock fracture signals. Subsequently, the U-Net model is trained to capture key features, achieving a bi-directional synergistic optimization of "physics-constrained and data-driven" approaches, thereby enhancing the accuracy of P-wave arrival picking.
We have revised Sections 2.1, 2.2, and 4.2 in the manuscript to further substantiate the aforementioned innovative claims. The specific modifications are highlighted in yellow. We thank the reviewer again for their insightful critique, which has helped us to clarify and solidify the theoretical contributions of this study.
Commented [IB3]: The architecture description lacks key information. Please give the number of layers, kernel sizes, activation functions, and training parameters such as learning rate, optimizer, and batch size. A schematic showing the data flow and feature dimensions would make the study easier to reproduce.
Response:
We thank you for your meticulous review and valuable comments on our research. In the U-Net variant model adopted in this study, we utilized a kernel size of 7. The ReLU activation function was selected for the entire network to ensure non-linear expression capabilities. Regarding the training parameters, we set the batch size to 32, the learning rate to and employed the Adam optimizer to adjust weight updates. These specific details have been supplemented and revised in Section 3.2 of the manuscript. Furthermore, to more clearly illustrate the data flow and the changes in feature dimensions, we have revised Figure 1 in the manuscript to include detailed annotations of the tensor shape transformation process between each layer.
Commented [IB4]: Although the model achieves good accuracy, statistical indicators such as confidence intervals or variance across repeated tests should be reported. Comparing the model against simpler baselines like STA/LTA or CNN-only architectures would help highlight the advantages of the proposed physics-constrained design.
Response:
We sincerely thank you for your concern regarding the rigor of the model evaluation. In this study, once all models were trained, they were used for inference on the test set with fixed parameters. Because the model architecture, the input data pre-processing pipeline, and the inference process itself contain no stochastic elements, the prediction results for the same test set are completely identical upon every execution. Therefore, repeated runs do not produce any fluctuation in the prediction results. This approach is standard practice within the current field of seismic AI. For example, prominent studies such as PhaseNet (Zhu & Beroza, 2019, GRL) and EQTransformer (Mousavi et al., 2020, Nat. Commun.) also reported the deterministic results of a single, optimal model on the test set, and did not provide statistical measures from repeated experiments.
Regarding the comparative methods, both AR-AIC and the Time-Window Energy Ratio (TWEFM) method are improvements upon the classic Short-Term Average / Long-Term Average (STA/LTA) algorithm. They are widely used for real-time arrival picking in seismic and microseismic applications within coal mine settings. Therefore, we selected AR-AIC and TWEFM for comparative analysis. The results demonstrate that the method proposed in this paper offers significant improvements in accuracy, operational efficiency, and other aspects.
Commented [IB5]: The proposed framework involves balancing multiple goals, including accuracy, noise robustness, and computational efficiency. Future studies could draw on systematic multi-objective optimisation frameworks to address these trade-offs more effectively. In particular, "Multi-objective optimisation of surveillance camera placement for bridge-ship collision early-warning using an improved non-dominated sorting genetic algorithm" provides a useful methodological reference for structured optimisation and could inspire parameter-tuning strategies in similar physics-constrained learning tasks.
Response:
We thank you for your valuable comments on this manuscript. Your suggestion regarding the introduction of a systematic multi-objective optimization framework to coordinate the trade-offs between accuracy, noise robustness, and computational efficiency is highly insightful. We fully recognize the importance of this direction. In future work, we will actively explore automatic tuning mechanisms for the intrinsic trade-off between model accuracy, noise robustness, and inference efficiency, aiming to achieve a synergistic improvement in both performance and efficiency. We have incorporated this discussion into the third paragraph of Section 6 in the revised manuscript. The specific modifications are highlighted in yellow on pages 18, lines 545-556.
Thank you once again for your meticulous review and constructive guidance.

Round 2
Reviewer 1 Report
Comments and Suggestions for Authors
I recommend acceptance of the manuscript in its current form. This study offers a valuable contribution to microseismic monitoring and signal processing in mining engineering, and it meets the publication standards of this journal.
Reviewer 2 Report
Comments and Suggestions for Authors
The raised questions/comments are replied.
Reviewer 3 Report
Comments and Suggestions for Authors
The authors corrected the data and obtained an improved result. The questions were answered, and overall, the results showed that noise is the main problem with this application of neural networks. The work is eligible for publication.